# Evaluation in a Cytokine Storm Model In Vivo of the Safety and Efficacy of Intravenous Administration of PRS CK STORM (Standardized Conditioned Medium Obtained by Coculture of Monocytes and Mesenchymal Stromal Cells)

**DOI:** 10.3390/biomedicines10051094

**Published:** 2022-05-08

**Authors:** Juan Pedro Lapuente, Gonzalo Gómez, Joaquín Marco-Brualla, Pablo Fernández, Paula Desportes, Jara Sanz, Mario García-Gil, Fernando Bermejo, Juan Víctor San Martín, Alicia Algaba, Juan Carlos De Gregorio, Daniel Lapuente, Almudena De Gregorio, Belén Lapuente, Sergio Gómez, María de las Viñas Andrés, Alberto Anel

**Affiliations:** 1R4T Molecular and Cell Biology Research Laboratories, Fuenlabrada Hospital, 28942 Madrid, Spain; gonzalog628@gmail.com (G.G.); pablo.fv@outlook.com (P.F.); jcdegregorio69@gmail.com (J.C.D.G.); daniel_lapuente_hernandez@yahoo.es (D.L.); almudenadegregorio@gmail.com (A.D.G.); belelapu@gmail.com (B.L.); sergocast@gmail.com (S.G.); v.andres@livingcells.org (M.d.l.V.A.); 2Group Immunity, Cancer and Stem Cells, Faculty of Sciences, University of Zaragoza, 50009 Zaragoza, Spain; joaquin_marco_91@hotmail.com; 3GMP Facility, Peaches Biotech, 28050 Madrid, Spain; paula_phisiup@hotmail.com (P.D.); j.sanz@iptam.es (J.S.); 4Pharmacy Department, Fuenlabrada Hospital, 28942 Madrid, Spain; mgarciagil@salud.madrid.org; 5Digestive Department, Fuenlabrada Hospital, 28942 Madrid, Spain; fernando.bermejo@salud.madrid.org; 6Medicine Department, University Rey Juan Carlos, 28942 Madrid, Spain; 7Internal Medicine Department, Fuenlabrada Hospital, 28942 Madrid, Spain; juanvictor.san@salud.madrid.org; 8Clinical Assay Department, Fuelabrada Hospital, 28942 Madrid, Spain; alicia_algaba@hotmail.com

**Keywords:** PRS CK STORM, cytokine storm, mesenchymal stem cells, M2 macrophages, coculture, crosstalk, secretome, inflammation

## Abstract

Our research group has been developing a series of biological drugs produced by coculture techniques with M2-polarized macrophages with different primary tissue cells and/or mesenchymal stromal cells (MSC), generally from fat, to produce anti-inflammatory and anti-fibrotic effects, avoiding the overexpression of pro-inflammatory cytokines by the innate immune system at a given time. One of these products is the drug PRS CK STORM, a medium conditioned by allogenic M2-polarized macrophages, from coculture, with those macrophages M2 with MSC from fat, whose composition, in vitro safety, and efficacy we studied. In the present work, we publish the results obtained in terms of safety (pharmacodynamics and pharmacokinetics) and efficacy of the intravenous application of this biological drug in a murine model of cytokine storm associated with severe infectious processes, including those associated with COVID-19. The results demonstrate the safety and high efficacy of PRS CK STORM as an intravenous drug to prevent and treat the cytokine storm associated with infectious processes, including COVID-19.

## 1. Introduction

The term cytokine storm refers to a profuse increase in cytokines, chemokines, growth factors, and interferons, causing severe inflammation, which, due to its severity, can be life-threatening because of multi-organ failure, or, at best, lead to irreparable tissue damage due to fibrosis [1,2,3].

Such cytokine storms associated with severe infectious processes represent a serious global problem, as this is a potentially life-threatening condition. To appreciate the seriousness of this condition, it is sufficient to recall that the current estimated annual incidence of septic processes, based on data from industrialized countries alone, is currently estimated at 48.9 million cases, killing an estimated 19.4 million people worldwide each year, thus accounting for 19.7% of global mortality worldwide [4,5].

Cytokine storms have been associated with a multitude of infectious and non-infectious diseases [6]. The association of cytokine storms with various pathologies, such as graft-versus-host disease [7], infections of various causative agents, such as bacteria [1], viruses [8] or fungi [9], autoimmune diseases [10], or even acute pancreatitis [11], has been reported. Special mention should be made of the cytokine storm associated with SARS-CoV-2 infection that caused so many deaths in the COVID-19 pandemic [12].

In the specific case of the cytokine storm associated with an infectious process, inflammation begins when macrophage-like cells of the innate immune system recognize pathogen-derived stimuli through their pattern-recognition receptors (PRRs) by recognizing structures called pathogen-associated molecular patterns (PAMPS) [13,14,15]. In addition to this stimulus, other stimuli will come from self-molecular structures derived from damage caused to tissues and cells, which are known as damage-associated molecular patterns (DAMPS) [16,17,18,19,20]. This whole process will generate a cascade of pro-inflammatory cytokine production whose main function is to regulate the duration and intensity of the immune response to the pathogen [21]. Thus, the cytokine storm is basically characterized by an exaggerated production of soluble pro-inflammatory and profibrotic mediators (especially IL-1β, IL-6, and TNF-α), together with an aberrant immunopathological reaction involving a lack of coordination between the innate and adaptive immune system with an overactivation of the innate immune system, the main cellular actors being macrophages, dendritic cells, monocytes, neutrophils, and T-lymphocytes [1,22,23,24]. Due to this cytokine storm, a situation of multi-organ hyperinflammation will be triggered, usually affecting mainly the lung and pancreas, among other organs, and often leading to acute respiratory distress syndrome (ARDS) and/or acute lung injury (ALI), which can result in multi-organ failure. However, although the association of increased levels of proinflammatory and profibrotic cytokines and chemokines with increased levels of morbidity and mortality following an infectious process is well known, we still do not have an adequate drug to treat the cytokine storm [25].

Given the variability we observed in our previous publication in in vitro tests, where we studied the safety and in vitro efficacy of our drug on a model to study the reactivity of PBMCs from different donors to LPS stimulation, we decided, in the present research, to use the standardized THP-1 cell line to repeat these tests prior to in vivo application.

Subsequently, based on the previous results of our group’s published in vitro research, intercellular communication between monocytes/macrophages and cells involved in tissue regeneration, such as mesenchymal stromal cells (MSCs) and primary tissue cells, is essential for tissue regeneration and recovery of homeostasis. Since this intercellular communication drives an anti-inflammatory immunomodulatory response in inflammation-resolving processes, we applied the monocyte–mesenchymal stem cell coculture secretome (hereafter PRS CK STORM) in a murine model of cytokine storm associated with severe infectious processes, such as severe COVID-19 or other infections, to firstly assess the safety of the drug and, secondly, its efficacy.

We believe that the application of PRS CK STORM, a standardized conditioned medium from a coculture of M2 macrophages with MSCs where, theoretically, all the growth factors, cytokines, and chemokines that are naturally produced by M2 macrophages and MSCs, associated with innate immunity, will be present, respecting natural pleiotropic relationships, with an immuno-modulatory cytokine profile expected to have a potent anti-inflammatory action, could be applied in the prevention and control of cytokine storms associated with infectious processes, such as those associated with COVID-19 or other types of infections.

## 2. Materials and Methods

### 2.1. Obtaining the Drug PRS CK STORM

For the development of the experiment, raw materials are needed to create the PRS CK STORM drug to be used on the animal model. The general process is summarized in Figure 1 and the detailed protocol of the manufacturing process is described in our latest paper, published in Biomolecules [26].

### 2.2. Mesenchymal Cell Culture

Adipose tissue samples were obtained from one healthy donor during routine abdominoplasty with the patient’s informed consent and were isolated, cultured, and characterized at Histocell’s advanced cell therapy factory in Bilbao. Cells were cultured in DMEM+ 1% penicillin/streptomycin (Sigma–Aldrich, St. Louis, MO, USA) +2.5% platelet lysate (UltraGro, AventaCell BioMedical Corp. Ltd., Atlanta, GA, USA) culture medium, changing medium every 4 days, performing the necessary passages until a homogeneous MSC population was obtained (usually at passage 3 so that, upon thawing of the cells, they remain at passage 4). After culture, cells were frozen on a freezing ramp of −0.5 °C/min to −80 °C in freezing medium consisting of 10% StemCellBanker (Amsbio, Abingdon, UK), following protocols previously described by Nieto-Aguilar et al. in 2011 [27] and Carriel et al. in 2013 [28]. They were grown to passage 4 and seeded in six-well plates at a density of 50,000 cells/well. An automatic counter, BioRad TC-20 (BioRad, Hercules, CA, USA) was used for counting.

### 2.3. Collection of Monocytes

The starting sample was a bag of whole blood from a donation at the Hospital Universitario de Fuenlabrada. It was diluted by half with physiological saline and a density gradient was performed with 20mL of Ficoll-Histopaque 1077 (Sigma–Aldrich, St. Louis, MO, USA) and 25 mL of the diluted blood. The bands obtained by centrifugation at 400× *g* for 30 min, without brake, were washed a total of four times with saline. Between the second and third wash, a lysis buffer was used to remove the erythrocytes still present in the sample, leaving it to act for 5 min. Finally, peripheral blood mononuclear cells (PBMCs) were seeded in 175 cm^2^ culture flasks at 3 × 10^8^ cells/flask with CTS AIM-V culture medium without phenol red (Gibco-BRL, Grand Island, NY, USA) supplemented with 1% stable glutamine (Dipeptiven^®^, Fresenius Kabi, Bad Homburg, Germany). After 2 h incubation at 37 °C and 5% CO_2_, monocytes were selected by adhering to the culture plastic and were obtained through cell scrapers after washing the surface with saline to remove lymphocytes present in the sample. They were seeded onto Transwell^®^ inserts (Falcon, PET, 1 μM pore size) (Corning, Corning, NY, USA) for six-well plates with a pore size of 1 μm at a density of 2 × 10^6^ cells/insert.

### 2.4. Coculture Monocytes/MSCs

Monocyte seedings were changed to monocyte culture medium (CTS-AIM-V™, Gibco-BRL, Waltham, MA, USA) with 2 mL per well, and empty inserts were placed in the wells, with a small amount of medium to adjust their porous membrane properties, and incubated again for 15 min. Finally, monocytes were seeded in a final volume of 1.5 mL and supplemented with M-CSF (R&D Systems, Minneapolis, MN, USA) at a final concentration of 10 ng/mL.

### 2.5. Collection and Conditioning of Supernatants

Every 3 to 4 days, supernatants were collected from the cocultures, centrifuged at 1800× *g* for 10 min and 4 °C, decanted to remove cell pellets, and stored at −80 °C. A total of six collections were performed, reaching 23 days of coculture. Once all partial samples were obtained, they were thawed at room temperature, mixed, and sterilized by filtration with 0.22 μm filters (Merck KGaA, Darmstadt, Germany). The final product was packaged in differentiated containers with the doses foreseen for its use. Part of the prepared doses was subjected to concentration, using Vivaspin tubes (Sartorius, Göttingen, Germany), capable of concentrating liquids. Centrifugation was carried out at 4 °C to maintain the properties of the biomolecules present for as long as necessary to achieve the desired concentration. In this way, the concentrated supernatant was adjusted to a 2.5× nd 5× concentration of the original product by diluting with the culture medium used (CTS-AIM-V™, Gibco-BRL, Waltham, MA, USA), which is the one used for PRS CK STORM.

### 2.6. Characterization of the Secretome

For quantification of the secretome of both cell types and coculture, 30 growth factors, cytokines, and chemokines were quantified using either ELISA or Multiplex assay (ProcartaPlex 45 PLEX, Invitrogen, Grand Island, NY, USA) strictly following the manufacturer’s instructions. A Luminex Labscan 100 plate reader (Luminex Corporation, Austin, TX, USA) was used for the determinations. The molecules quantified by Multiplex were the following: MIP1-α, SD-1α, IL-27, LIF, IL-2, IL-4, IL-5, IP-10, IL-6, IL-7, IL-8, IL-10, PIGF-1, Eotaxin, IL-12p70, IL-13, IL-17A, IL-31, IL-1Ra, SCF, RANTES, IFN-γ, GM-CSF, TNF-α, HGF, MIP-1β, IFN-c, MCP-1, IL-9, VEGF-D, TNF-β, NGF-β, BDNF, GRO-α, IL-1α, IL-23, IL15, IL-18, IL-21, FGF-2, IL-22, PDGF-BB, VEGF-A, TIMP-1, and MMP-3. For quantification of IGF-1, TIMP-1, and MMP1, a double sandwich ELISA technique was used according to the manufacturer’s instructions (DuoSet ELISA kit, R&D, Minneapolis, MN, USA) and quantification was determined using an iMark plate reader (BioRad, Hercules, CA, USA) and absorbances were measured at 450 and 570 nm.

### 2.7. THP-1 In Vitro Inflammation Model

The THP-1 cell line was cultured and further differentiated into macrophages to generate in vitro models of biosafety and efficacy. THP-1 monocytic cells (CellLineService, cat. No.: 300356) were cultured and expanded using RPMI 1640 (Lonza, Basile, Switzerland) supplemented with 10% fetal bovine serum (FBS) (Corning, NY, USA), 1% penicillin/streptomycin (FBS) (Corning, NY, USA), and 10% penicillin/streptomycin (FBS) (Corning, New York, USA), 1% penicillin/streptomycin (P/S) (Lonza, Basile, Switzerland), 1 mM sodium pyruvate (Lonza, Basile, Switzerland), and 1% MEM non-essential amino acids (Gibco, Thermo Fisher Scientific, Waltham, MA, USA), henceforth THP-1 medium. Cells were maintained at a density of 1 × 10^6^ cells/mL to ensure adequate growth and a stable phenotype. Next, 48 h prior to LPS stimuli, cells were differentiated into resting macrophages using Phorbol 12-Mystyrate 13-Acetate (PMA) (Sigma Aldrich, Saint Louis, MO, USA) at 5 ng/mL in THP-1 medium, as described in the protocol used by Park et al. in 2007 [29]. After this differentiation process, the cells were used for our experiments. All cell cultures were maintained at 37 °C in an atmosphere of 5% CO_2_ and 98% relative humidity.

First, to test in vitro safety, an MTT-type assay was performed on THP-1 cells transformed to macrophages by adapting the method of Chen et al. in 2016 [30]. Differentiated THP-1 cells were washed three times with 0.2 mL of tempered THP-1 medium without PMA and allowed to rest for 30 min before LPS stimuli. After cell media change, cells were treated with 10 ng/mL LPS (Sigma–Aldrich, Burlington, MA, USA) in RPMI 1640 medium and treated with 100 μL of PRS CK STORM or the control, as defined above, by adding tempered THP-1 medium to obtain 200 μL of cell culture per well. Three wells were seeded with THP-1m cells only, three were seeded with THP-1m cells stimulated with LPS at a concentration of 10 ng/mL, three were seeded with THP-1m cells stimulated with LPS at a concentration of 10 ng/mL and PRS CK STORM at a low dose, three were seeded with THP-1m cells stimulated with LPS at a concentration of 10 ng/mL and PRS CK STORM at a medium dose, three were seeded with THP-1m cells stimulated with LPS at a concentration of 10 ng/mL and PRS CK STORM at high dose and, finally, as controls, three were seeded with THP-1m cells stimulated with LPS at a concentration of 10 ng/mL and hydrocortisone at 10 μg/mL (Sigma–Aldrich, Burlington, MA, USA). The different increasing concentrations of PRS CK STORM were calculated as a function of TIMP-1 content in the conditioned medium (low at 594.86 pg total TIMP-1, medium at 1189.72 pg TIMP-1, and high at 5948.6 pg TIMP-1). After incubation of all cultures for 96 h, 10 μL/well of an aqueous solution (5mg/mL) of tetrazolium blue (Sigma–Aldrich, Burlington, MA, USA) was added. The MTT/tetrazolium blue solution was incubated for 4 h at 37 °C, 5% CO_2_; after incubation, the plates are centrifuged at 600× *g* for 7 min to precipitate the cells and formazan crystals, and after removal of the medium, the formazan crystals are solubilized by adding 200 μL/well of DMSO. The plates are incubated at 37 °C for 10 min and shaken at 250 rpm using a plate shaker (JP Selecta, Abrera, Catalonia, Spain). Results are obtained by measuring the absorbance of each well at 570 nm on an iMark plate reader (BioRad, Hercules, CA, USA).

The in vitro inflammation model was then generated by differentiating 1 × 10^6^ cells/mL THP-1 cells in exponential growth phase to resting macrophages in 12-well plates (Nunc, Thermo Fisher, Waltham, MA, USA) (final volume 1ml) and after 48 h pretreatment with PMA, THP-1m cells were washed three times with 0.5 mL of tempered THP-1 medium without PMA and allowed to incubate for 30 min before LPS stimuli. Once resting, cells were treated with 10 ng/mL LPS (Sigma–Aldrich, Saint Louis, MO, USA) in RPMI 1640 medium and PRS CK STORM at a low dose, as this was the dose at which the product had shown the best in vitro biosafety profile. The medium was pre-warmed to room temperature or tempered THP-1 medium. The final volume of each well was 1 mL with a THP-1m cell seeding density of 1 × 10^6^ cells/mL (three wells). Three wells seeded in the same way and with the same cell density but treated with hydrocortisone at 10 μg/mL (Sigma–Aldrich, Saint Louis, MO, USA) were used as controls. After 5 h of stimulation, the supernatants were collected and frozen at −80 °C for cytokine analysis, in which variations in the secretion of TNF-α and IL-1 β, as main mediators of the response to lipopolysaccharide, were studied. All experimental conditions were assayed in triplicate to ensure sufficient statistical robustness.

### 2.8. In Vivo Safety and Efficacy Assay

First, to test the safety (pharmacodynamics and pharmacokinetics) of PRS CK STORM, a total of 15 Balb/c male mice were used as an animal model. This experiment was performed at the animal facility of Biomedical Research Centre of Aragon (CIBA, Zaragoza, Spain) under the European recommendations on animal ethics and approved by the Ethics Committee for Animal Experimentation from the research center (Annex 1). Mice were stabled in groups of five for at least one week before the experiment, maintained at a constant temperature and humidity and 12-h day/night cycle, with access to food and water ad libitum. Mice were divided into three groups of five animals, each of which was treated with a different dose (low, medium, or high) and different number of administrations from single dose to five doses o.d. (See explanatory Table 1).

Work schedule for animal experimentation was as follows:Day 0 (0 h in the graphics): Before PRS CK STORM administration, around 300 µL of whole blood were extracted from the submandibular vein of all mice. From these samples, 7/15 were randomly chosen (# 16, 17, 22, 23, 25, 26 and 27), from the rest, 8/15 (# 18, 19, 20, 21, 24, 28, 29 and 30) were chosen and separated for further analysis; samples 7/15 were chosen for biochemical determination and samples 8/15 for cytokine analysis.Day 1 (0 h): It was decided to separate blood collection from PRS CK STORM administration into separate days, to not distress the animals too much. As both day 0 and 1 are time points right before PRS CK STORM injections, both are considered as “0 h”. All mice were administered a volume of 100 µL PRS CK STORM intraperitoneally (IP), at their corresponding doses (Dose n°1).Day 2 (24 h): The first mouse from each group was sacrificed (n°16, 21, and 26). At least 700 µL of whole blood were extracted from these mice by cardiac puncture. The rest of the animals (four mice per group) were administered PRS CK STORM by IP injection, at their corresponding doses (Dose n°2).Day 3 (48 h): The second mouse from each group was sacrificed (n°17, 22, and 27). At least 700 µL of whole blood were extracted from these mice by cardiac puncture. The rest of the animals (three mice per group) were administered PRS CK STORM by IP injection, at their corresponding doses (Dose n°3).Day 4 (120 h): The third mouse from each group was sacrificed (n°18, 23, and 28). At least 700 µL of whole blood were extracted from these mice by cardiac puncture. The rest of the animals (two mice per group) were administered PRS CK STORM by IP injection, at their corresponding doses (Dose n°4).Day 5 (144 h): The fourth mouse from each group was sacrificed (n°19, 24, and 29). At least 700 µL of whole blood were extracted from these mice by cardiac puncture. The rest of the animals (one mice per group) were administered PRS CK STORM by IP injection, at their corresponding doses (Dose n°5).Day 6 (168 h): The last mouse from each group was sacrificed (n°20, 25, and 30). At least 700 µL of whole blood were extracted from these mice by cardiac puncture.

The administration of the first three doses was consecutive and 72 h elapsed between doses 3 and 4. Indeed, administrations 4 and 5 were resumed at 120 and 144 h, respectively, from the first administration (day 1, 0 h). In addition, all animals who received the dose 5 administration were sacrificed at time 168 h (24 h after last administration). Table 2 shows a summary of the distribution of animals, dose, and number of administrations when they were sacrificed:

Every day, after blood sample obtention, serum was extracted, which was used as material for the next analysis. Blood-derived serum samples were used for the following analyses:Safety analysis—The following biochemical parameters were determined: levels of biliary acid, albumin, alanine aminotransferase (ALT), bilirubin, cholesterol, alkaline phosphatase, γ-glutamyl transferase, and blood urea nitrogen were measured using a VetScan VS2 Chemistry Analyzer (Comprehensive Diagnostic Profile protocol #500-0038 Abaxis, Union City, CA, USA) (Annex 2) on all samples from day 1–6 and, of 7/15 samples from day 0, randomly chosen samples (#16, 17, 22, 23, 25, 26, and 27). This analysis was also performed at the animal facility in CIBA.Pharmacodynamic analysis—the presence of several murine cytokines was quantified. HGF, TNF-α, IL-12 p70, IL-1β, IL-6, IL-10, IFN-γ, and TIMP-1 were determined in all samples collected from all animals on day 1–6 (see Table 2) and in 8/15 samples from day 0, randomly chosen (#18, 19, 20, 21, 24, 28, 29, and 30). Determinations were decided by MULTIPLEX (Mouse Premixed Multi-Analyte Kit; Catalog Number LXSAMSM) assays in Annex 3 at the Cellular Sorting and Cytometry Department in CIBA.Pharmacokinetic analysis—the presence of several human cytokines was quantified: HGF, TNF-α, IL-12 p70, IL-1β, IL-6, IL-10, IFN-γ, and IL-1RA in all samples from day 1–6 (24 h after last IMP administration). Determinations were set by MULTIPLEX (Human Premixed Multi-Analyte Kit; Catalog Number LXSAHM) assays in Annex 4, also at the Cellular Sorting and Cytometry Department in CIBA.

For the second experiment, the efficacy test, a total of 35 C57/BL6 mice were used. All mice experimentation was performed at the animal facility of Biomedical Research Centre of Aragón (CIBA, Zaragoza, Spain), under the European recommendations on animal ethics and as previously approved by the Ethics Committee for Animal Experimentation from the research center (see Annex 5). Mice were stabled in groups of five for at least one week before the experiment and maintained at constant temperature and humidity and 12-h day/night cycle, with access to food and water ad libitum.

To perform the experimental model of acute lung damage and inflammation due to sepsis, the experimental model described by Stephens et al. in 2015 [31] was used. For this model, 8–10-week-old female and male C57BL/6 mice were administered 5 mg/kg of bacterial lipopolysaccharide (LPS) (Sigma L4130, Sigma–Aldrich, Saint Louis, MO, USA) in 50 μL of physiological solution via the retro-orbital route. This type of LPS administration is necessary because intravenous or intraperitoneal injection of LPS, which is often accompanied by high plasma concentrations of inflammatory cytokines, hypotension, and hypothermia (Lewis et al., 2016) [32], causes survival in mice to decrease steeply after 24 h due to sepsis shock, especially in the case of intravenous injection (Fang et al., 2018; Starr et al., 2010) [33,34]. For this technique, 27.5-gauge insulin needles and 0.5-inch syringes Terumo U-100 (Shibuya-ku, Tokyo, Japan) were used, and it is recommended that the volumes injected do not exceed 150 μL. Since the needle was placed in the retrobulbar space, mice had to be anaesthetized with inhalational isoflurane and a drop of ophthalmic anesthetic (0.5% proparacaine hydrochloride ophthalmic solution, Alcon Laboratories, Fribourg, Switzerland) in the eye that received the injection. To reduce the potential distress of the mice due to LPS, buprenorphine hydrochloride was administered in water at the established dose of 0.056mg/mL. Treatments were administered daily for four consecutive days, and they consisted of:The same PRS CK STORM batch used in the safety in vivo experiment was used in this efficacy experiment, and its full characterization can be found in Appendix A. After characterization at Living Cells laboratory (University Hospital of Fuenlabrada, Madrid, Spain), this batch was frozen in a −80 °C freezer and shipped in dry ice before being used in this experiment. The original batch showed a TIMP-1 concentration of 59,485.27 pg/mL. After centrifugation in Vivaspin tubes of 15 mL (12,400 MW) for 80 min, 3600 *g* at 4 °C, a 5× concentration was reached, meaning there were 297,426 pg/mL. Since the results of the safety study showed that PRS CK STORM was well tolerated in the three tested doses administered intravenously, the high dose of this study (5948 pg TIMP-1) was chosen as the mean dose for this in vivo efficacy study, with two additional doses tested, one lower (mean/2, 2380 pg TIMP-1) and one higher (mean × 2, 11,897 pg TIMP-1). These doses were prepared by diluting the high dose at 1/2 or 1/4, respectively, in phosphate buffer saline (PBS). See in Table 1 the calculation of TIMP-1 absolute doses for administered volume of 40 µL. In all cases, PRS CK STORM was injected intravenously. To avoid premature over-inflammation by LPS, the first dose was injected 24 h prior to LPS administration as a pre-treatment, with the next treatment injected on the day of model generation and every 24 h thereafter.

The election of these doses was based on the characterization of the batch used of PRS CK STORM manufactured for investigational purposes and considering both the cytokine composition and the acceptable biological activity observed in vitro. The three doses were selected for in vivo safety testing in mice based on absolute TIMP-1 amount (considered as anti-inflammatory cytokine with a quantitatively relevant present in PRS CK STORM). In all cases, PRS CK STORM was obtained in Living Cells laboratory (University Hospital of Fuenlabrada, Madrid, Spain) from conditioned medium from human allogenic M2-macrophages, cocultured ex vivo with human adipose tissue-derived mesenchymal stem cells. The full characterization of the PRS CK STORM batch can be found in Appendix A. The original batch was used as the “high dose” in this experiment. The “medium dose” and “low dose” were prepared by diluting the high dose at 1/2 or 1/4, respectively, in phosphate buffer saline (PBS).

Drug vehicle: consists of the media culture where PRS CK STORM was collected, which will be referred to as “monocyte medium” (CTS-AIM-V™, Gibco-BRL, Waltham, MA, USA) and was also injected intravenously.Gold Standard: this treatment consists of a classic drug used against the inflammation process, Kineret^®^ (Anakinra, or IL-1RA antagonist, Swedish Orphan Viovitrum AB, Stockholm, Sweden) at 149.25 mg/mL, administered orally, with the help of an intragastric probe.

In the experimental design, 35 mice were divided in 7 groups of 5 animals, each of them receiving different treatments, as explained in Table 3 below:

Work schedule for animal experimentation was performed every morning at 9 a.m., and was as follows:Day −1: 24 h before LPS injection, corresponding mice were administered monocyte medium, PRS CK STORM (low, medium, or high dose), or gold standard. As explained above, this pre-treatment was performed to try to prevent premature over-inflammation by LPS (Dose n°1).Day 0: Each group was injected with their respective treatment (monocyte medium, PRS low, PRS medium, PRS high, none, or gold standard) (Dose n°2). Immediately after, corresponding mice were retro-orbitally injected with LPS. To mitigate discomfort derived from LPS, a dose of buprenorphine hydrochloride (Buprecare^®^, Leonvet, León, Spain) was administered subcutaneously (0.15 mg/mouse in 0.1 mL) in those mice after LPS injection.Day 1: Around 200 µL of whole blood were extracted from the submandibular vein of all mice, and serum was obtained from them before administering the treatment. Animals in groups carrying LPS were also administered buprenorphine hydrochloride and their respective treatment (monocyte medium, PRS low, PRS medium, PRS high, none, or gold standard) (Dose n°3).Day 2: As on the day before, around 200 µL of whole blood were extracted from the submandibular vein of all mice, except for the Control group, and serum was obtained from them, before administering the treatment. Animals in groups carrying LPS were also administered buprenorphine hydrochloride and their respective treatment (monocyte medium, PRS low, PRS medium, PRS high, none, or gold standard) (Dose n°4).Day 3: All mice were sacrificed. Whole blood was extracted by cardiac puncture (500–800 µL) and serum was obtained from them. Additionally, necropsies were performed, and liver, heart, spleen, lungs, and kidneys were collected from the animals. Half of every organ was frozen and kept at −80 °C, while the other half was preserved in formaldehyde (4% *v*/*v* in sterile water).

During and after the end of the experiment, mice and the samples collected from them were used for the following analyses:

#### 2.8.1. Efficacy Evaluations

Irwin tests and temperature measurements: mice were regularly supervised to detect any possible alteration in their behavior or comfort derived from treatments and/or handling. Based on the manuscript from (Mathiasen and Moser, 2018) [35], an Irwin test was confectioned in a table and used as a template. Therefore, the Irwin test was performed every day of the experimental procedure on all mice, immediately before treatment administrations. Temperatures of mice were measured by rectal probe at three different time points: immediately before LPS injections, 20 min after LPS injections, and 48 h after LPS administration. 

Time from CO_2_ exposure to death: when mice were sacrificed on day 3, animal’s resistance to CO_2_ exposure until their death was classified in three clear groups: less than 30 s, 30–60 s, and more than 60 s.

Macroscopic observations of organs at necropsy: after animal sacrifices and blood extraction, necropsies were performed and relevant findings in organs were detected and photographed (Annex 6).

Histological study: Both frozen (−80 °C) and formaldehyde-preserved organs were sent to Living Cells’ laboratory for histopathological study (Annex 7).

#### 2.8.2. Efficacy Evaluations

In the safety analysis, the following biochemical parameters were determined: levels of albumin, alkaline phosphatase, alanine aminotransferase, amylase, bilirubin, blood urea nitrogen, creatinine, globulin, total protein, glucose, calcium, phosphorus, sodium, and potassium were measured using a VetScan VS2 Chemistry Analyzer (Comprehensive Diagnostic Profile protocol #500-0038 Abaxis, Union City, CA, USA) (Annex 2) from all samples beginning on day 3, excepting some animals who prematurely died before the end of the experiment. This analysis was also performed at the animal facility in CIBA.

#### 2.8.3. Pharmacodynamic Analysis

The presence of several murine cytokines was quantified. TNF-α, IL-1β, IL-6, IL-10, TIMP-1, and MMP-3 levels were determined by Multiplex assays (MILLIPLEX MAP Cat: MCYTOMAG-70K, Merck KGaA, Darmstadt, Germany) (Annex 8–10) at the Cellular Sorting and Cytometry Department in CIBA. TNF-α, IL-1β, IL-6, and IL-10 were measured in all samples from day 1 to day 3, but TIMP-1 and MMP-3 were only measured in some samples from day 3 due to serum volume limitation.

#### 2.8.4. Pharmacokinetic Analysis

The presence of several human cytokines from our IMP was quantified. IFN-α2, IL-1β, IL-1RA, IL-6, and TNF-α levels were also determined by Multiplex assays (Human Cytokine/Chemokine/Growth Factor Panel a Magnetic Bead Panel Cat. # HCYTA-60K, Merck KGaA, Darmstadt, Germany) (Annex 11) at the Cellular Sorting and Cytometry Department in CIBA. Once again, owing to the lack of samples, especially from days 1 and 2, this analysis was only performed with some of the samples from day 3.

### 2.9. Statistics

The MTT and cytokine release assays as well as the cytokine analysis of the culture supernatants and the different biochemical parameters were subjected to statistical analysis. Statistical analysis was performed with Excel (Microsoft, Albuquerque, New Mexico, USA). All statistics were calculated with data from independent experiments performed in triplicate. An ANOVA test was performed to determine statistically significant differences between the experimental groups studied using GraphPad Prism software version 8.4.0 for Mac OS X (GraphPad Software, San Diego, CA, USA) to perform the calculations. The level of statistical significance was set at *p* < 0.05.

## 3. Results

### 3.1. Characterization of the PRS CK STORM

The compositional analysis of PRS CK STORM demonstrates a clear anti-inflammatory and immunomodulatory profile against a potential cytokine storm, as shown in Table 4, which shows the results of the comparative cytokine profile between three different production batches of the PRS CK STORM.

The complete batch characterization that was manufactured is shown in Table 5.

### 3.2. Differentiation of THP-1 Cells to Macrophages

The cells after 48 h of stimulation with PMA (5 ng/mL) gain adherence to the culture media and take on a morphology like macrophages. After 24 h of culture, the number of cells in suspension decreases and the number of cells adherent to the plastic increases, a sign of successful differentiation. After 48 h, about 90% of the cells are adherent to the plastic and are used in the experimental model.

### 3.3. In Vitro MTT and Bioactivity Assays on THP-1-Derived Macrophages

First, the results of the MTT assay were analyzed with THP-1 cells previously stimulated with PMA for 48 h to promote their adhesion to the plastic and the expression of innate and helper receptors. The results are shown in Figure 2.

Secondly, once the low dose was selected as a possible therapeutic dose for the in vivo experiment, the results of the in vitro inflammation model performed on THP-1 cells previously stimulated with PMA for 48 h to promote their adhesion to the plastic and the expression of innate and auxiliary receptors, and stimulated with LPS, were analyzed. The results are shown in Figure 3.

As can be seen in Figure 3, the cells are sensitive to LPS stimuli and the PRS CK STORM drug studied demonstrates a potent anti-inflammatory effect, especially at high doses, with a dose that demonstrates anti-inflammatory potency such as that shown by the soluble hydrocortisone used as a control.

### 3.4. In Vivo Safety Test 

We treated a total of 15 male Balb/c mice with three increasing doses of PRS CK STORM to test the safety of the investigational product.

#### 3.4.1. Animal Behavior and Comfort

Throughout the course of the experiment, mice were supervised to guarantee their wellness and assess the alterations that might occur following PRS CK STORM administration. The Score Panel can be found in Table 6 and is interpreted as follows: every aspect gathered in the panel has a given number (first column). If any mouse presents one of those characteristics, it is scored with that number. The total amount is calculated and, depending on the score, a conclusion can be extracted (see Total Score by the end of the panel).

#### 3.4.2. Biochemical Determinations

Biochemical determinations (safety analysis) are shown in Figure 4. Consequently, no statistical differences were calculated. Exact values of the profile are shown in Appendix A.

#### 3.4.3. Cytokine Analysis

The effect of PRS CK STORM treatment on several relevant murine cytokines (pharmacodynamic effect) and human cytokines (pharmacokinetic effect) was assessed by multiplex assays. As already explained in the Materials and Methods section, eight of the fifteen day 0 samples were randomly selected as baseline values for murine cytokine measurements (protocol in Annex 3). For human cytokines (pharma; as treatment had not yet started, the assay was discarded at that experimental (pre-dose) point (protocol in Annex 4), these two assays were performed at the Department of Cell Sorting and Cytometry, CIBA. Apart from the *t* = 0 h samples, and as in the safety analysis, the mice were distributed so that there was one mouse at each time point and dose concentration (*n* = 1), so no statistical differences were calculated. From these values, graphs were made for murine (Figure 5) and human (Figure 6) cytokines.

### 3.5. In Vivo Efficacy Test 

The results of the Irwin test [35] were analyzed to evaluate the overall effect of the conditioned medium administered intravenously to mice in which the cytokine storm model was generated by retroorbital injected LPS (see Appendix A). In the basal control and PBS groups, all animals scored on all items as standard. In the LPS control group mouse 11 died, and mice 12, 13, 14 and 15 scored the items stopping, piloerection and body temperature as slight increase, the item dehydration as moderate increase and the items exploratory activity, contact reaction and extremities tone as moderate decrease. In the control group LPS + Gold Standard mouse 18 died, and mice 16, 17, 19 and 20 scored the items stopping, tremors and diarrhea as slight increase, the items piloerection and dehydration as moderate increase and the items exploratory activity, contact reaction and extremities tone as moderate decrease. In the low dose LPS + PRS CK STORM control group all mice scored in the items stop, piloerection, diarrhea, and dehydration as slightly increased. In the control group LPS + PRS CK STORM at intermediate doses, mouse 27 died and mice 26, 28, 29 and 30 scored in the items stopping, diarrhea and dehydration as slightly increased. In the control group LPS + PRS CK STORM at high doses, mouse 32 died and mice 31, 33, 34 and 35 scored in the items stopping, diarrhea and dehydration as slightly increased. The mean pre-treatment temperature in all mice was 38.1 °C. In the LPS-treated groups, the mean temperature dropped 20 min after LPS injection to a mean temperature of 34.2 °C in all LPS-treated groups, but 48 h after the start of the respective treatments in all groups, the mean temperature dropped to 30.2 °C on average in the Gold Standard-treated group, increased to 35.4 °C on average in the low-dose PRS CK STORM-treated group, increased to 35.1 °C on average in the intermediate-dose PRS CK STORM-treated group, and finally increased to 35.2 °C on average in the high-dose PRS CK STORM-treated group.

From the blood obtained after euthanasia and exsanguination of the animals, biochemical tests were carried out to determine the biochemical profiles, which are shown in Figure 7. The Mann–Whitney test was used for statistical significance (* *p* < 0.1; ** *p* < 0.05).

Figure 8 shows the evolution of the values obtained for murine cytokines (TNF-α, IL-1β, IL-6, MMP-3, IL-10, and TIMP-1) throughout the treatment.

Figure 9 shows the evolution of the values obtained for human cytokines (IFN-α2, IL-10, IL-1β, IL-6, TNF-α, and IL-1Ra) throughout the treatment.

Finally, histopathological analysis of samples obtained from various organs of the mice after necropsy showed patchy interstitial thickening of the lung in most of the sample in the LPS treatment, whereas, as the concentration of the test drug (PRS CK STORM) increased, the observed damage reversed to the point that there was no lung damage at all in those treated with high doses. Slight liver and spleen damage was observed, which the drugs also reversed. As for heart and kidney, no pathological findings were detected (images not shown). Table 7 shows the information obtained from the different histological sections.

Figure 10 shows examples of the lung sections studied in the different groups of the experiment.

## 4. Discussion

The PRS CK STORM test drug was produced by non-direct contact coculture of M2 macrophages with MSCs at Histocell’s advanced cell therapy factory.

Multiplex characterization of PRS CK STORM batches for cytokines, chemokines, and growth factors shows that all those with a pro-inflammatory role are below detection limits [40], except IL-6 and IL-8, which are considered dual cytokines with respect to the immune response, i.e., depending on the context in which they act, they can be either pro- or anti-inflammatory [41]. Several authors have shown that to define the pro- or anti-inflammatory character of a mixture of cytokines and growth factors, the most important thing is to define the ratios between them [42,43,44,45]. In this regard, the analysis of the PRS CK STORM ratios shown in Table 1 shows a clear anti-inflammatory profile, with the ratio of pro-inflammatory (IL-1, IL-6, IL-18, TNF-α, IFN-γ, IL-17) versus anti-inflammatory (TIMP-1, IGF-1, IL-10, IL-1RA) cytokines being 0.00042.

In vitro tests to demonstrate potential efficacy (bioactivity test) and safety (MTT type test) are considered of great importance for the future development of this drug, as similar products successfully employ a strategy of mixing conditioned media from different donors to obtain one or more homogeneous batches from very heterogeneous components [46] and will allow the establishment of quality and safety conditions for both the manufacture and release of the product. The THP-1 cell line used in this study does not originally show sensitivity to 100 ng/mL LPS, according to our own experience (data not shown) and descriptions in the literature [29]. These cells gain sensitivity to LPS when differentiated to macrophages with the addition of PMA to the culture medium. Upon differentiation, they gain macrophage (THP-1m) characteristics and begin to express increased levels of membrane receptors capable of detecting LPS (data not shown). This cell line allows us to study the responses of a cell type very similar to tissue-resident human macrophages, not present in the PBMC population, with very little variability and a very stable phenotype. Using the THP-1 cell line [47] compared to the PBMCs used by our group in another study is a great advantage in the method, since, as can be seen in the results, although IL-1β responses vary and are lower in THP-1 compared to PBMCs, TNF-α responses remain almost identical, with practically the same release, avoiding possible cross-reactions of eosinophils or mast cells present in PBMCs that can alter the levels of IL-1β and/or TNF-α upwards. The study of these inflammatory mediators and not others related to the immune response is because these are the main and first factors secreted by PBMCs when they are stimulated in an innate immunity reaction in the first instance to a stimulus from a pathogen. These factors, IL-1β and TNF-α, have a wide range of functions from cell proliferation to induction of apoptosis [48]. They are also at the end of the signaling pathways used to generate inflammation in our model; IL-1β is secreted when the inflammasome is engaged and caspase 1 converts inactive pro-IL-1β into bioactive and secretable IL-1β, while TNF-α is released from the cell membrane when the nuclear factor kappa-beta (NF-κβ) pathway is activated [49]. These two cytokines, although not the only ones responsible for inflammation, are able to provide a good insight into the inflammatory response derived from the LPS stimulus as they are the main factors studied by most research groups performing this type of modelling [50,51,52].

The results shown in Figure 2 of the safety MTT assay demonstrate that any of the doses of PRS CK STORM can be used in the in vivo assay, since all of them show a statistically significant reduction in the reduction of formazan, although it is the low dose of PRS CK STORM that seems to show a higher level of reduction. It was therefore decided to use this dose to test possible in vitro efficacy, a result shown in Figure 3.

In respect to the in vivo safety assay, the Irwin test (Table 6) shows that four out of fifteen mice manifested a slight increase in their resistance to handling and agitation before the beginning of the treatment. As the procedure was underway, this behavior disappeared, and no mice presented any type of alterations until the end of the experiment. These observations suggested that PRS CK STORM treatment does not induce any secondary effects, at least at a macroscopic level, at the experimental doses used. In the same experiment, it was observed that the moderate dose induced an increase in alkaline phosphatase after the first injection, which was maintained in the second dose, but no relationship between the dose level or the number of administrations with the increase was evident, and its concentration returned to normal values at the end of treatment (time = 0 h). A peak in standard alkaline phosphatase values after the fourth injection was also observed when using PRS CK STORM at low and high doses; however, they also returned to normal after the fifth injection and the final values remained normal (Figure 4). As for alanine aminotransferase, a single increase (four times above standard values) was detected in a single mouse after the third injection of high dose PRS CK STORM. Despite this, the observed increase returned to the normal range quickly and was maintained for the remainder of the treatment (Figure 4), again ruling out possible dose dependence. Finally, no variations were found in the concentration of γ-Glutamyl Transferase; therefore, this enzyme does not appear to be affected by PRS CK STORM administration (Figure 4). Therefore, from these data, it can be concluded that PRS CK STORM does not affect the liver, as the single peaks detected appear to be isolated and not dose dependent. Moreover, in all these cases, either after the first two doses or after the time interval (48–120 h), all mice quickly recover their standard values and, at the end of the experiment, even in mice injected five times with this drug, all values normalize. Bile acid did appear to increase after the first injection at the moderate doses and the third injection at the other two concentrations; however, as before, these are single peaks that resolved the following day and, at the end of the experiment, remained in the standard range (Figure 4). Bilirubin rose slightly after the first (low dose) or second (moderate and high dose) administration, but again, the increases quickly returned to normal and remained low until the end (Figure 4). Blood urea nitrogen showed no variation throughout the experiment (Figure 4). As for albumin, which is an acute-phase negative protein, it did not change at the low dose of PRS CK STORM, but an increase could be detected after the second injection in the other two concentrations (Figure 4). Regardless of this, the mice normalized their values at the end. In the case of cholesterol, there was a moderate increase, but it did not seem to be too relevant for the health and well-being of the mice (Figure 4). As for murine cytokines (pharmacodynamic results), starting with anti-inflammatory cytokines, more specifically HGF and TIMP-1, they were not indifferent to PRS CK STORM treatment, as an increase and subsequent stabilization was detected (Figure 5). This was especially clear in the case of TIMP-1, where an increase in the order of 2.5 was observed after administration of the high dose of PRS CK STORM (only at the high dose, after the second injection). Despite this, the values of both proteins decreased after the 72-h untreated interval and, at the endpoint, normalized (Figure 5). Taken together, this could be relevant, as it suggests that the mice may initiate an anti-inflammatory and reparative mechanism that could resolve the increase in transaminases. As for the remaining anti-inflammatory cytokine, IL-10, there was little variation throughout the experiment (Figure 5). IL-1β was unchanged from day 0 values (there was even a decrease in the fourth treatment when the high dose of PRS CK STORM was used, but it returned to standard levels) (Figure 5). The same interpretation can be applied to IL-12 p70 and IL-6 (the apparent increase in IL-6 observed on the last day at the highest dose was still within baseline values) (Figure 5). Notably, IFN-γ and TNF-α, both of which are considered pro-inflammatory or loss of homeostasis markers, along with IL-1β, were always below the limit of detection and did not increase after PRS CK STORM treatment (Figure 5). As for the human cytokines (pharmacodynamic results), it was evident that the concentration of TNF-α was barely detectable at baseline and remained virtually unchanged throughout the experiment (Figure 6), coinciding with the baseline levels of PRS CK STORM (Appendix A). The same was true for HGF and IL-6 (Figure 6), whose concentrations were lower than baseline PRS CK STORM (Appendix A) and their murine counterparts (Figure 5). In the case of IFN-γ, IL-10, and IL-1β, their concentrations were very low and, in most samples, were below the limit of detection. Thus, despite being apparently higher than their PRS CK STORM characterization reference values (see both Figure 6 and Table 5), their concentrations were almost undetectable and presumably unaltered. As for IL-12 p70, although it appeared to increase for all doses, they eventually resolved and declined to baseline concentrations, suggesting that this cytokine is eventually cleared from the body without further complications (Figure 6). Finally, in the case of the anti-inflammatory IL-1RA receptor, although an unusual peak was initially observed at all doses, its concentration decreased greatly from day 1 of treatment, falling below baseline values by the end of the experiment (Figure 6).

In respect to the in vivo efficacy assay, the experimental model employed uses LPS as the causative agent of acute lung damage, causing a cytokine storm in the body of the mice such as that produced by any lung infection, as, for example, occurs in COVID-19 disease. Irwin’s test (see Appendix A) showed a statistically significant reduction in body temperature in LPS-treated mice. In fact, the mean basal temperature went from approximately 36.3 °C to 24 °C after LPS stimulation. This necessitated placing all animals in a thermal blanket to preserve life. This was not performed in the Gold Standard group, which may have contributed to the lower mean temperature of the group. In all groups treated with LPS, there was a slight increase in stooping, slight diarrhea, and consequent dehydration of the animals, which was mild in all groups treated with PRS CK STORM and more intense in the group treated with Anakinra. On the other hand, a slight increase in piloerection was observed in the group treated with low-dose PRS CK STORM and in the group treated with LPS alone, which was greater in the group treated with Anakinra. Mild tremor and a significant decrease in exploratory activity as well as general reactivity were observed in both the LPS-only and Anakinra-treated groups. These were not observed in any of the PRS CK STORM-treated groups. Of the 35 animals in the study, 4 of them died at some intermediate point, as shown in Appendix A. On the day after administration, mouse 12 and 32 died and were found to have severe gastritis at necropsy. On the second day after administration, mouse 27 (necropsy showed clear signs of lung inflammation with severe gastritis) and mouse 32 died (necropsy showed slight signs of lung inflammation). Irwin’s test showed that PRS CK STORM markedly attenuated the detrimental effects of the cytokine storm associated with LPS administration, which is like that produced during a severe acute phase of an infectious process, as occurs in COVID-19. Biochemical profiles of mice and rats are poorly described in the literature and vary widely between sexes and strains, as well as between different sources consulted, which complicates data analysis. Most of the proteins and metabolites analyzed follow the same trend, irrespective of the experimental group observed. As for albumin (Figure 7C), the main protein present in the blood, there is a decrease in albumin in all groups administered LPS, which fits with what was described by Ballmer et al. in 1994 [53], as hypoalbuminemia occurs when the organism undergoes sepsis due to infection. Related to this is the decrease in total protein (Figure 7A), as a decrease in albumin will lead to a decrease in total protein because it is at very high concentrations. The decrease observed for glucose (Figure 7K) could be explained by the presence of IGF-1 in the drug administered, which emulates insulin and increases glucose uptake by tissues [54], as well as the presence of other sugars in the secretome that make up the PRS CK STORM that could interact, decreasing basal glucose levels. The increase in globulin (Figure 7N) can also be explained by LPS, as these proteins increase in inflammatory situations. In addition, within the large group of globulins, the growth of immunoglobulins in the presence of LPS is noteworthy [55]. The rest of the parameters remain without significant modifications between the different treatments (Figure 7B,D–J), suggesting that they are all within the normal range. For potassium (Figure 7M), its concentration could not be determined because they all exceeded the detection limit of the program used, so it is not possible to determine whether its value is modified by inducing LPS and drug treatment.

As for pharmacodynamic and pharmacokinetic results, the levels of the following cytokines, both murine (TNF-α, IL-1β, IL-6, MMP-3, IL-10, and TIMP-1) and human (IFN-α2, IL-10, IL-1β, IL-6, TNF-α, and IL1Ra), were monitored to assess the effect of the PRS drug CK STORM, which demonstrated its therapeutic potential by inhibiting the production of pro-inflammatory cytokines while increasing the production of anti-inflammatory cytokines such as IL-10, whose function lies in its ability to inhibit the synthesis of these pro-inflammatory cytokines. Figure 8 shows the evolution of these murine cytokines detected in the sera of the mice on each of the days of treatment. In view of these results, it can be deduced that stimulation with bacterial lipopolysaccharide (LPS) can induce the expected inflammatory response, being more pronounced in the most acute phase, the day after the administration of the treatment, and decreasing over time. In the control group and the vehicle, no pro-inflammatory cytokines were observed, confirming that LPS is the cause of this response. In the group treated with Anakinra (Gold Standard), a significant decrease in the concentration of all the murine pro-inflammatory cytokines studied (TNF-α, IL-1β, IL-6) was observed. It should be noted here that mouse number 32, which died on day 2 of the experiment, recorded very high levels of IL-1β on day 1 (42.53 pg/mL), which could justify its death one day later. This is striking, as Anakinra is an interleukin-1 (IL-1) receptor antagonist. Similarly striking in Figure 9 are the elevated levels of human IL-1β and IL-1Ra recorded in mouse 31 of the same group on the day of the experiment. In this case, the IL-1Ra levels could be explained by the same treatment received (Anakinra), as the same increase was observed in mouse 33, but in the latter, the IL-1β levels on day 3 were low. The opposite occurred in mouse 31, where a large increase in human IL-1β was observed. This could be due to a cross-reactivity phenomenon. Treatment with PRS CK STORM at the different concentrations tested also causes a considerable decrease in murine pro-inflammatory cytokine levels over time, especially at the high dose, as can be seen in Figure 8. Although, in general, it appears that both the high dose PRS CK STORM treatment and the gold standard Anakinra treatment seem to control the cytokine storm; in the high dose PRS CK STORM group the mean murine TIMP-1 level on day 3 of the experiment was 24,551.55 pg/mL, while the same mean level in the Anakinra group was only 9441.99 pg/mL, with the mean murine MMP-3 values in both groups being very similar. This could be related to the anatomopathological observations (see Table 7 and Figure 10) observed in these groups, as the Anakinra-treated group showed lesions in the lung, liver, and spleen that were very similar to those observed in the LPS-only group, reporting inflammatory infiltrates thickening the alveolar interstitium in 50% of the lung section as well as mild inflammatory lesions in both liver and spleen, while no lung lesions were observed in the group treated with high dose PRS CK STORM (further information can be found in Annex 6 and 7 of Appendix A). Another important observation in this regard, which histopathology confirmed a posteriori, was the fact that the seconds it took for the mice to die in the CO_2_ chamber were timed (an average of 118 sg for the high dose PRS CK STORM-treated mice, versus an average of 27 sg for the Anakinra-treated mice). TIMP-1 is an inhibitory molecule that regulates matrix metalloproteinases (MMPs) and disintegrin metalloproteinases (ADAM and ADAMTS) [56], down-regulating metalloproteinases (MMPs) and thus playing a crucial role in the composition of the extracellular matrix (ECM), favoring tissue regeneration, and slowing fibrotic scarring processes [57]. On the other hand, it is also noteworthy that in both the group treated with high dose PRS CK STORM and the group treated with Anakinra, IL-1β levels decreased on day 3 to the same mean levels (6.1 pg/mL). It is clear that with Anakinra being an IL-1 receptor antagonist, this decrease is a direct consequence of its mechanism of action [58,59], but in the case of high dose PRS CK STORM, the IL-1Ra content is much lower than in the case of Anakinra (7849.47 pg total versus 5 × 10^9^ pg total, respectively), so the anti-inflammatory effects observed by the action of PRS CK STORM may not be due to the direct action of its IL-1Ra content, but are possibly due to a combined action of the set of molecules contained in it on different points of various metabolic cascades linked to pattern recognition receptors (PRRs), producing cytoprotective, anti-apoptotic and tissue regenerative effects, and downregulating the production of fibrosis [46,60,61]. Our group is currently immersed in the study of the mechanism of action of PRS CK STORM, as well as in the toxicology tests under GMP conditions, necessary to bring the investigational product (IP) to human clinical trials.

Therefore, it appears that high dose PRS CK STORM may be useful in preventing and treating the cytokine storm associated with severe infectious processes, such as those associated with COVID-19, since the data analyzed suggest that at this dose it could prevent the associated life-threatening aggravation while avoiding the appearance of fibrosis in the tissues affected by the inflammation.

## 5. Conclusions

The results show that PRS CK STORM is composed of a secretome with a clear anti-inflammatory, anti-fibrotic, and regenerative profile. The use of THP-1m cells in the methods explored in the present study as tests for examining the biosafety and anti-inflammatory potential of PRS CK STORM makes these tests reproducible and reliable, and they should be implemented as quality control tools in the production processes of different conditioned media. PRS CK STORM at high doses has demonstrated a high capacity, not only to reduce acute inflammation in cytokine storms associated with severe infections by immunoregulating the activity of the innate immune system and improving its coordination with adaptive immunity, but also in its efficacy as an anti-fibrotic drug, avoiding the medium and long-term negative consequences of acute inflammatory phenomena.

Taken together, the results suggest that high dose PRS CK STORM secretome from coculture of M2 macrophages with MSCs may become a safe and effective biological drug to prevent and treat the cytokine storm associated with severe infectious processes, such as that associated with COVID-19.

## 6. Patents

This research work has resulted in the patent PCT/EP2020/059365 “Composition for tissue regeneration, method of production and uses thereof”.

## Figures and Tables

**Figure 1 biomedicines-10-01094-f001:**
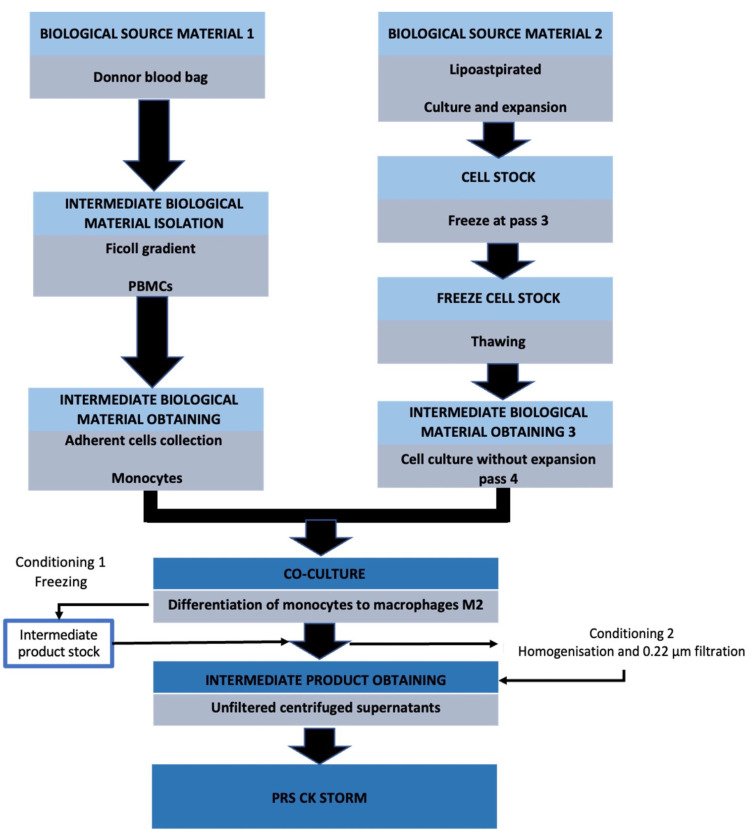
Schematic description of the production process of allogeneic conditioned medium derived from M2-type macrophages and enriched with MSCs [26].

**Figure 2 biomedicines-10-01094-f002:**
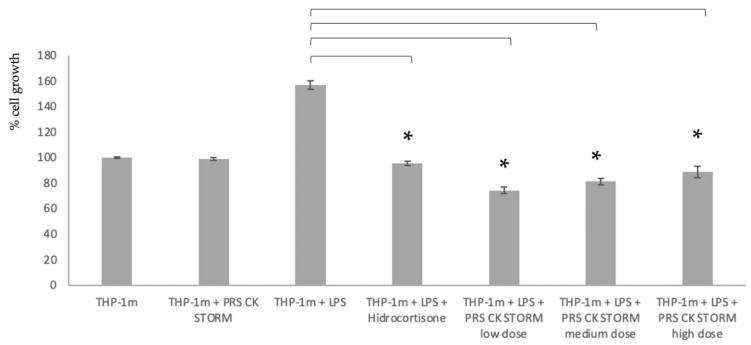
MTT assay with macrophages derived from the THP-1(THP-1m) line, and these were analyzed in three independent experiments with two replicates of the analytical technique. Error bars indicate the standard deviation between samples. Asterisks represent a *p*-value < 0.05.

**Figure 3 biomedicines-10-01094-f003:**
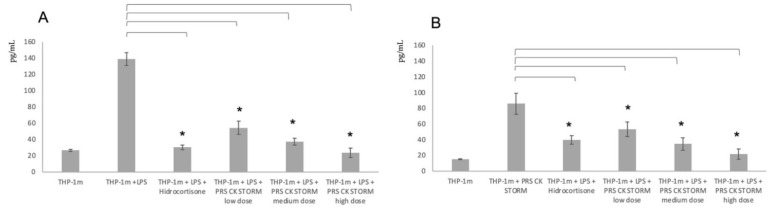
Release of pro-inflammatory cytokines from THP-1 cells differentiated to macrophages (THP-1m) in the in vitro model of inflammation. THP-1m were stimulated with LPS in the same way as PBMCs used in previous studies. The graphs show the pg/mL of each of the factors studied ((**A**) = IL-1β; (**B**) = TNF-α). Error bars correspond to the standard deviation of 3 independent experiments analyzed in duplicate. * represent a *p*-value < 0.05.

**Figure 4 biomedicines-10-01094-f004:**
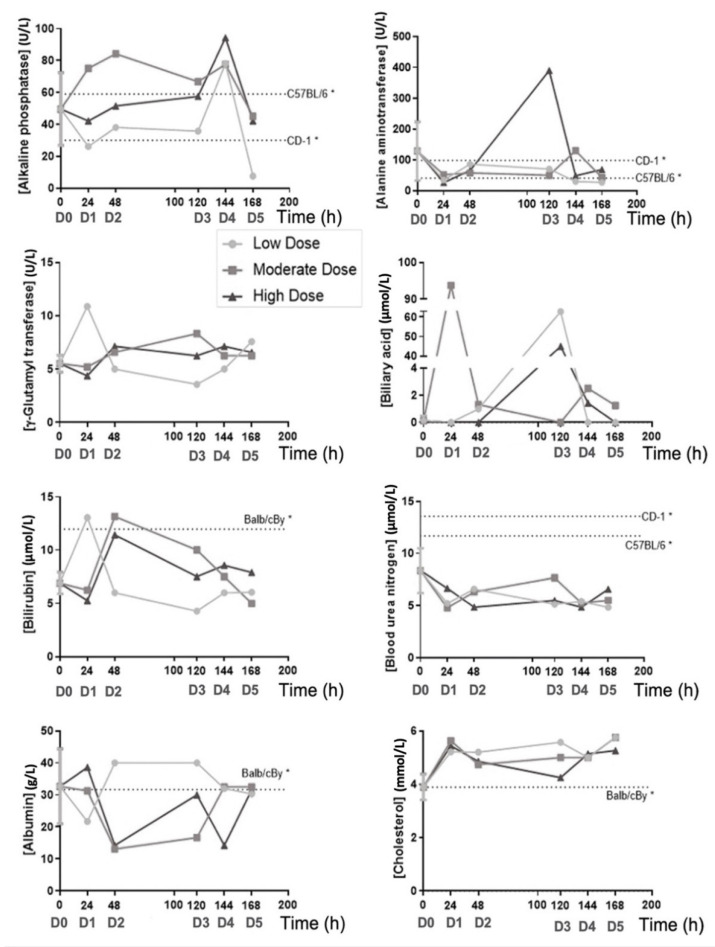
Biochemical profile of liver enzymes. Blood samples were extracted from 7 out of 15 mice from day 0 (before treatment) and from each one of them at their respective experimental point (see Methodology). Afterwards, their sera were used to quantify the levels of Biliary Acid, Albumin, Alanine Aminotransferase, Bilirubin, Cholesterol, Alkaline Phosphatase, γ-Glutamyl Transferase, and Blood Urea Nitrogen using a VetScan VS2 Chemistry Analyzer. Time = 0 h is represented as the mean +/−SD from seven samples. D0–5 indicates the number of doses already administered to those mice at every time point. * Reference data were extracted from: Laboratory Animal Medicine, 3rd Edition (Elsevier, 2015) [36,37,38,39].

**Figure 5 biomedicines-10-01094-f005:**
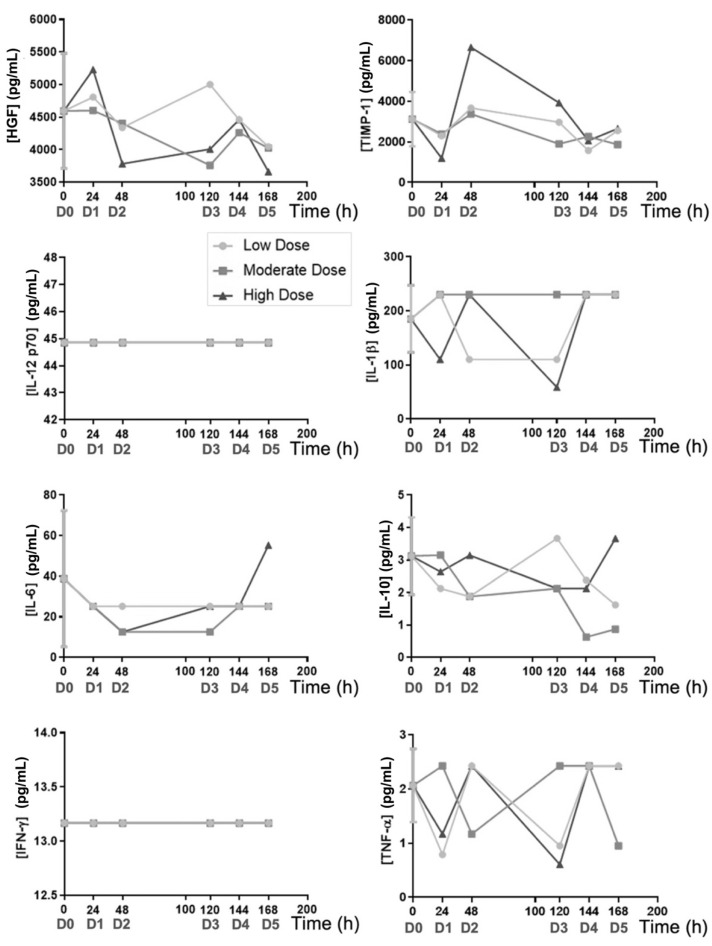
Analysis of murine cytokines. Blood samples were extracted from 8 out of 15 mice from day 0 (before treatment) and from each one of them at their respective experimental point (see Methodology). Afterwards, their sera were used to quantify the levels of human TNF-α, IL-6, IL-10, IL-1β, IFN-γ, IL-1Ra, IL-12 p70, and HGF, by multiplex assays. D0–5 indicates the number of doses already administered to those mice at every time point. Time = 0 h is represented as the mean +/− SD from eight samples. Appendix A is available in Appendix A: Values of murine cytokines analyzed by multiplex assay.

**Figure 6 biomedicines-10-01094-f006:**
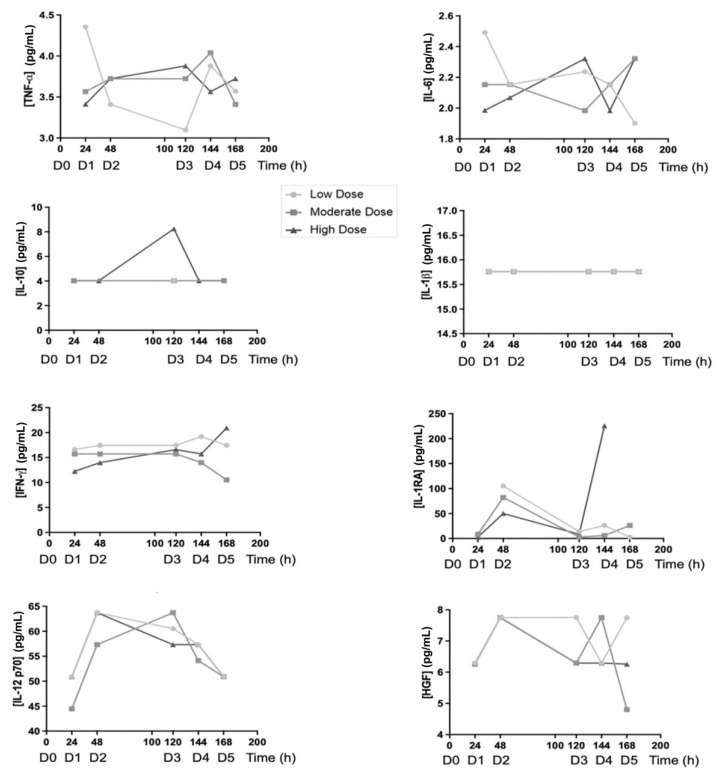
Analysis of human cytokines. Blood samples were extracted from 8 out of 15 mice from day 0 (before treatment) and from each one of them at their respective experimental point (see Methodology). Afterwards, their sera were used to quantify the levels of mouse HGF, TNF-α, IL-12 p70, IL-1β, IL-6, IL-10, IFN-γ, and TIMP-1 by multiplex assays. D0–5 indicates the number of doses already administered to those mice at every time point. Time = 0 h is represented as the mean +/− SD from 8 samples. Appendix A is available in Appendix A: Values of human cytokines analyzed by multiplex assay.

**Figure 7 biomedicines-10-01094-f007:**
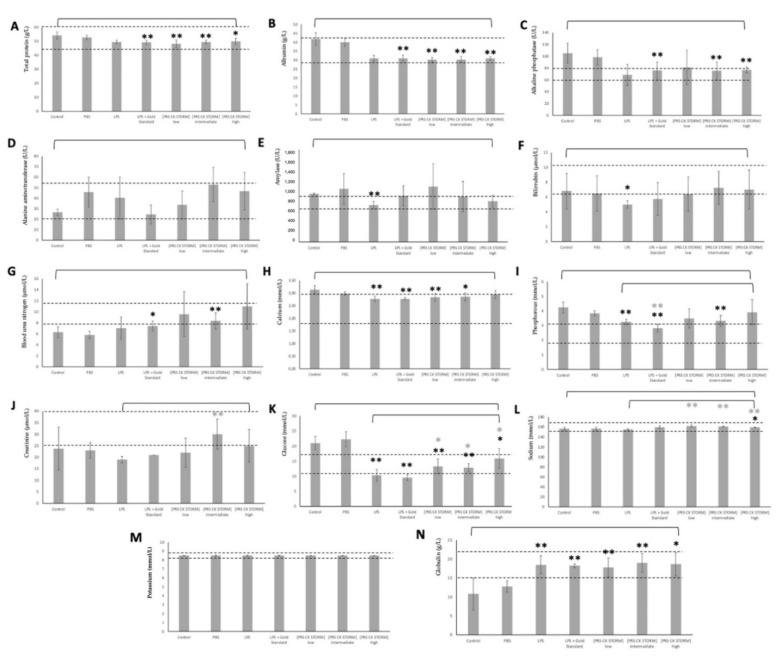
Analysis of biochemical parameters represented as the mean +/− SD from 8 samples. Blood samples were extracted from 8 out of 15 mice from day 0 (before treatment) and from each one of them at their respective experimental point (see Methodology). (**A**) shows the values of total protein in g/L, (**B**) shows the values of albumin in g/L, (**C**) shows the values of alkaline phosphatase in U/L, (**D**) shows the values of alanine aminotransferase in U/L, (**E**) shows the values of amylase in U/L, (**F**) shows the values of bilirrubin in μmol/L, (**G**) shows the values of blood urea nitrogen in μmol/L, (**H**) shows the values of calcium in mmol/L, (**I**) shows the values of phosphorous in mmol/L, (**J**) shows the values of creatinine in μmol/L, (**K**) shows the values of glucose in mmol/L, (**L**) shows the values of sodium in mmol/L, (**M**) shows the values of potassium in mmol/L, and (**N**) shows the values of globulin in g/L. The Mann-Whitney test was used for statistical significance (* *p* < 0.1; ** *p* < 0.05), comparing the control group with the other groups (dark black asterisks), and the same test between the LPS group and the groups treated with either PRS CK STORM or Gold Standard (light grey asterisks).

**Figure 8 biomedicines-10-01094-f008:**
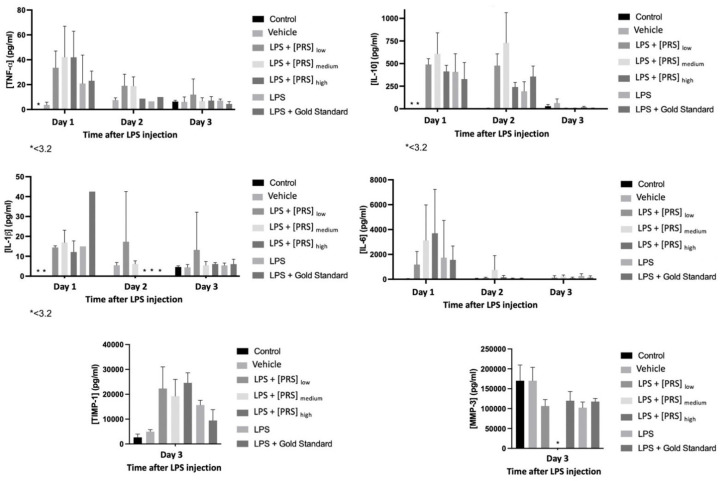
Serum values of the different murine cytokines 1, 2, and 3 days after treatment expressed in pg/mL as the mean of the values of the 5 mice in each of the experimental groups. Evolution of murine cytokines. Asterisks (*) indicate a concentration below 3.2 pg/mL, the detection limit of the assay.

**Figure 9 biomedicines-10-01094-f009:**
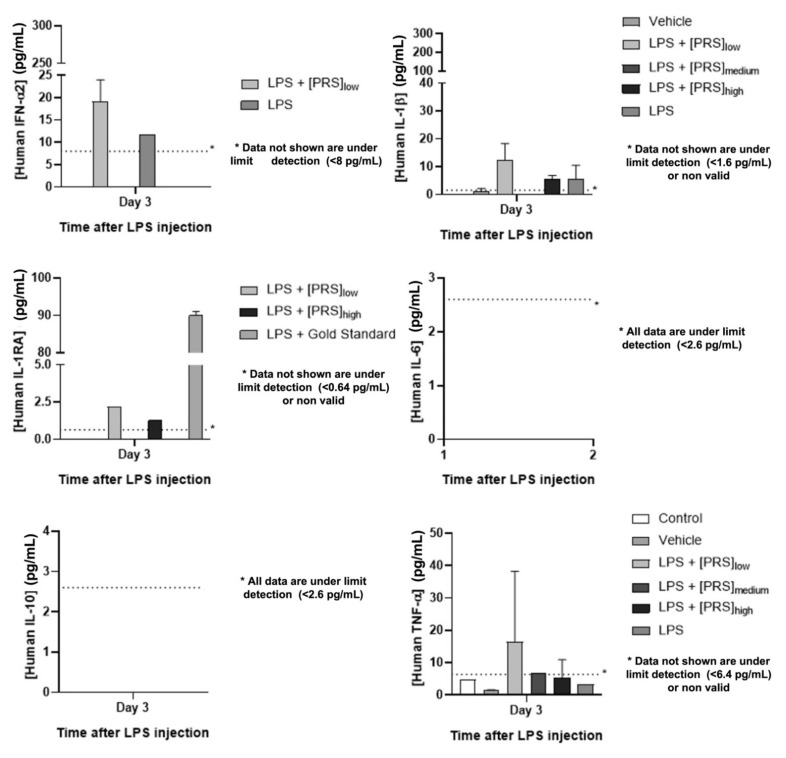
Serum values of the human different cytokines 3 days after treatment expressed in pg/mL as the mean of the values of the 5 mice in each of the experimental groups. Evolution of murine cytokines. Asterisks (*) indicate a concentration below the detection limit of the assay.

**Figure 10 biomedicines-10-01094-f010:**
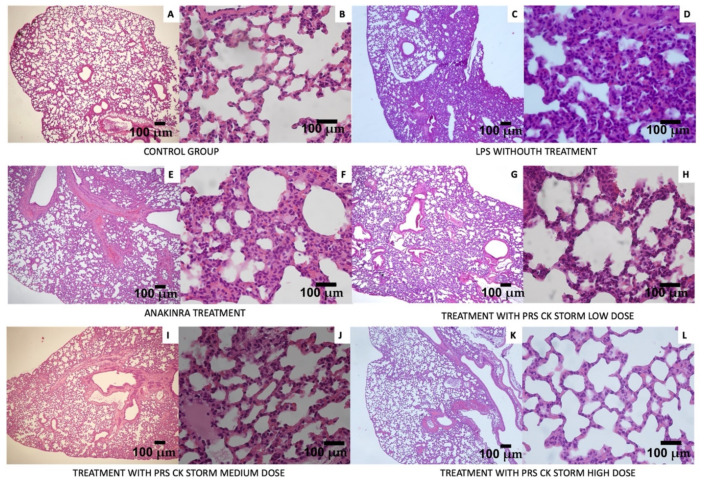
Lung anatomo-pathological study in the different groups of mice in the experiment (all figures shows representative microscopic images of the lung). In the group treated with the highest dose of our conditioned medium, the appearance of the lung is very similar to that of the untreated control group. (**A**,**B**) shows that no pathological findings were observed in the lung samples analyzed in the control group. (**C**,**D**) shows an irregular thickness of the interstitium in more than 50% of the section, and inflammatory cells can be observed as part of the interstitial thickness in the lung samples analyzed in the LPS group. (**E**,**F**) shows an irregular thickness of the interstitium in approximately 50% of the section, and inflammatory cells can be observed as part of the interstitial thickness in the lung samples analyzed in the LPS + Anakinra treated group. (**G**,**H**) shows that there are patchy inflammatory infiltrates thickening the interstices between the alveoli, which cover approximately 30% of the slice area in the lung samples analyzed in the low-dose LPS + PRS CK STORM treated group. (**I**,**J**) shows some patchy inflammatory infiltrates thickening the interstices between the alveoli, covering approximately a quarter of the slice surface in the lung samples analyzed in the group treated with intermediate doses of LPS + PRS CK STORM. (**K**,**L**) shows that no pathological findings were observed in the lung samples analyzed in the group treated with high doses of LPS + PRS CK STORM.

**Table 1 biomedicines-10-01094-t001:** Distribution of animals, dose, time, and number of administrations when they were sacrificed. (Time from hour 0 when the first administration was performed. In all cases, mice were sacrificed 24 h after its last administration).

Dose Time at Sacrifice	1 Adm. 24 h	2 Adm. 48 h	3 Adm. 120 h	4 Adm. 144 h	5 Adm. 168 h
Low	#16	#17	#18	#19	#20
Medium	#21	#22	#23	#24	#25
High	#26	#27	#28	#29	#30
n	3	3	3	3	3

**Table 2 biomedicines-10-01094-t002:** Description of 3 groups of animals and doses administered using 5 representative cytokines/growth factors.

Group	Number of Animals	#	Dose	Dose (pg/mL +/−20%)	Absolute Administered Dose (pg +/−20%)
1	5	16–20	Low	IL-1Ra (14,998 pg/mL) TIMP (14,871 pg/mL) IL-10 (0.06 pg/mL) HGF (211 pg/mL) IGF-1 (419 pg/mL)	IL-1Ra (1499 pg) TIMP (1487 pg) IL-10 (0.006 pg) HGF (21 pg) IGF-1 (42 pg)
2	5	21–25	Medium	IL-1Ra (28,996 pg/mL) TIMP (29,743 pg/mL) IL-10 (0.12 pg/mL) HGF (423 pg/mL) IGF-1 (838 pg/mL)	IL-1Ra (2899 pg) TIMP (2974 pg) IL-10 (0.012 pg) HGF (42 pg) IGF-1 (84 pg)
3	5	26–30	High	IL-1Ra (57,992 pg/mL) TIMP (59,485 pg/mL) IL-10 (0.24 pg/mL) HGF (845 pg/mL) IGF-1 (1675 pg/mL)	IL-1Ra (5799 pg) TIMP (5948 pg) IL-10 (0.024 pg) HGF (84 pg) IGF-1 (167 pg)

**Table 3 biomedicines-10-01094-t003:** Description of the groups of animals, LPS, and administered treatment. In the case of PRS CK STORM, 5 representative cytokines/growth factors were used.

Group	Number of Animals	#	Group Name	Treatment (Daily for 4 Days)	Volume Injected (µL)	Absolute Administered Dose +/−20%	LPS (5 mg/kg) (Yes/No)
1	5	1–5	Control	None	-	-	No
2	5	6–10	Drug vehicle	Monocyte medium	40	-	No
3	5	11–15	LPS + PRS _low_	PRS _low_		TIMP-1 (2380 pg)	Yes
4	5	16–20	LPS + PRS _medium_	PRS _medium_	40		Yes
5	5	21–25	LPS + PRS _high_	PRS _high_	40	TIMP-1 (5948 pg)	Yes
6	5	26–30	LPS	None	40	TIMP-1 (11,897 pg)	Yes
7	5	31–35	LPS + Gold Standard	Kineret^®^	-	-	

**Table 4 biomedicines-10-01094-t004:** Batch of PRS CK STORM anti-inflammatory (data expressed as the sum of the percentages representing the sum of the amounts of each of the named cytokines in pg/mL with respect to the total sum of all the cytokines studied in their composition).

Batch	CK Anti-Inflammatory (TIMP-1, IGF-1, IL-10, IL-1Ra)	CK Pro-Inflammatory (IL-1, IL-6, IL-18, TNF-α, IFN-γ, IL-17)
PRS CK STORM	99.420%	0.041%

**Table 5 biomedicines-10-01094-t005:** Mean values of the molecules studied. Values are shown in picograms per milliliter.

**MIP-1α**	**SDF-1α**	**IL-27**	**LIF**	**IL-1β**	**IL-2**	**IL-4**	**IL-5**
81.32 (SD 14.56)	247.56 (SD 46.78)	< 21.41	20.78 (SD 1.23)	<2.16	<7.21	<10.49	<9.90
**IP-10**	**IL-6**	**IL-7**	**IL-8**	**IL-10**	**PIGF-1**	**Eotaxin**	**IL-12 p70**
13.87 (SD 1.67)	403.78 (SD 34.56)	<0.99	268.45 (SD 29.81)	1.99 (SD 0.56)	<1.71	2.56 (SD 0.34)	<4.71
**IL-13**	**IL-17A**	**IL-31**	**IL-1Ra**	**SCF**	**RANTES**	**IFN-** **γ**	**GM-CSF**
<3.58	<2.27	<9.21	63,389.96	<3.58	<2.27	<9.21	63,389.96
**TNF-α**	**HGF**	**MIP-1β**	**IFN-α**	**MCP-1**	**IL-9**	**VEGF-D**	**TNF-β**
12.89 (SD 0.78)	371.56 (SD 54.67)	132.87 (SD 23.12)	<0.45	2876.34 (SD 345.12)	<2.89	<0.79	<5.69
**NGF-β**	**EGF**	**BDNF**	**GRO-α**	**IL-1α**	**IL-23**	**IL-15**	**IL-18**
<6.14	<1.78	<0.34	10.21	<0.61	<6.14	<1.78	<0.34
**IL-21**	**FGF-2**	**IL-22**	**PDGF-BB**	**VEGF-A**	**TIMP-1**	**MMP-3**	**MMP-1**
<6.37	<2.72	<18.07	14.87	<6.37	<2.72	<18.07	14.87

**Table 6 biomedicines-10-01094-t006:** Supervision Score Panel for mice used in animal experimentation.

	Mouse Supervision Protocol Score Panel			
Animal Number ID		16–19, 21–22, 25–29 (Before Treatment)	20, 23, 24, 30 (Before Treatment)	16–30 (During Treatment)
Body condition index	SCORE			
Normal	0	0	0	0
15–20% decrease	10			
>20% decrease (**)	19			
Response to handling				
Normal	0	0		0
Slightly increased/decreased	2		2	
Very increased/decreased, aggressive	5			
Behavior and appearance				
Aggressiveness towards other animals	1	0	1	0
Stereotypies	3	0	0	0
Piloerection	3	0	0	0
Piloerection and slight nose bleeding	5	0	0	0
Hunched back	10	0	0	0
Convulsions, diarrhea, coma (**)	19	0	0	0
Self-mutilations (**)	19	0	0	0
Severe respiratory distress (**)	19	0	0	0
Significant blood loss >20% (**)	19	0	0	0
Severe dehydration (**)	19	0	0	0
Blood extraction zone				
Slight inflammation	7	0	0	0
Necrosis/ulcer lesion (**)	19	0	0	0
TOTAL SCORE		0	3	0

Score: 0–4: normal; 5–9: Increase frequency of animal revisions; 10–18: Consult to veterinarian; >18: Mandatory sacrifice (**).

**Table 7 biomedicines-10-01094-t007:** Summary of the main events observed in the histological sections of the different organs to be studied.

Group	LUNG	HEART	LIVER	KIDNEY	SPLEEN
Control Group	1 *	1 *	1 *	1 *	1 *
LPS Group	2 *	1 *	6 *	1 *	9 *
Anakinra Group	3 *	1 *	7 *	1 *	9 *
PRS CK STORM_Low_ Dose	4 *	1 *	8 *	1 *	1 *
PRS CK STORM_Medium_ Dose	5 *	1 *	7 *	1 *	1 *
PRS CK STORM_High_ Dose	1 *	1 *	1 *	1 *	1 *

1 *: Image compatible with normality; 2 *: Inflammatory infiltrates that thicken the alveolar interstices in more than 60% of the histological section; 3 *: Inflammatory infiltrates that thicken the alveolar interstices in more than 50% of the histological section; 4 *: Inflammatory infiltrates that thicken the alveolar interstices in 30% of the histological section; 5 *: Inflammatory infiltrates that thicken the alveolar interstices in 20% of the histological section; 6 *: Clear dilatation of spaces between hepatic sinusoids; 7 *: Slight dilatation of spaces between hepatic sinusoids; 8 *: Central lobule vein with dilated sinusoidal capillaries; 9 *: Slight disorganization of the splenic pulps.

## Data Availability

The datasets generated and/or analyzed during the current study are available from the corresponding authors on reasonable request.

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
