# Peer review of "Evaluation in a Cytokine Storm Model In Vivo of the Safety and Efficacy of Intravenous Administration of PRS CK STORM (Standardized Conditioned Medium Obtained by Coculture of Monocytes and Mesenchymal Stromal Cells)"

_biomedicines, 2022, doi:10.3390/biomedicines10051094_

Round 1

Reviewer 1 Report

Dear Authors,

Congratulations for great work, very interesting and time consuming work. I only found minor errors or missed the given information - I included some suggestions to explain something a bit more.

In general, my opinion is to accept with minor corrections.

Good luck!

JG 

Author Response

Dear Colleague:

Thank you very much for the interest, work and time you have invested in reviewing our work. We deeply appreciate your help and comments.

Kind regards from Spain

Reviewer 2 Report

This is a well written  manuscript with some interesting findings, explored about the efficacy of intravenous PRS CK STORM  drug, which can prevent and treat cytokine storms associated with infectious diseases of any type, including COVID-19. Methodology and data executions were adequate. I would like to congratulate the authors on their wonderful work.

Author Response

(The authors gave the same response as above.)
